



# Mineral Dust Influence on the Glacial Nitrate Record from the RICE Ice Core, West Antarctica and Environmental Implications

Abhijith U. Venugopal[1], Nancy A.N. Bertler[1,2], Rebecca L. Pyne[2], Helle A. Kjær[3], V. Holly L. Winton[1], Paul A. Mayewski[4], Giuseppe Cortese[2]

[1]Antartic Research Centre, Victoria University of Wellington, Wellington, New Zealand.

[2]GNS Science, Lower Hutt, Wellington, New Zealand

[3]Centre for Ice and Climate, Physics of Ice, Climate and Environment, Niels Bohr Institute, University of Copenhagen, Denmark

[4]Climate Change Institute, University of Maine, Orono, ME, USA

Corresponding author: Abhijith Ulayottil Venugopal (abhi.ulayottilvenugopal@vuw.ac.nz)

**Abstract.** Nitrate ($NO_3^-$), an abundant aerosol in polar snow, is a complex environmental proxy to interpret owing to the variety of its sources and its susceptibility to post-depositional processes. During the last glacial period, when the dust level in the Antarctic atmosphere was higher than today by a factor up to ~25, mineral dust appears to have a stabilizing effect on the $NO_3^-$

concentration. However, the exact mechanism remains unclear. Here, we present new and highly resolved records of $NO_3^-$ and non-sea salt calcium ($nssCa^{2+}$, a proxy for mineral dust) from the Roosevelt Island Climate Evolution (RICE) ice core for the period 26 - 40 kilo years Before Present (ka BP). This interval includes seven millennial-scale Antarctic Isotope Maxima (AIM) events, against the background of a glacial climate state. We observe a significant correlation between $NO_3^-$ and $nssCa^{2+}$ over this period and especially during AIM events. We put our observation into a spatial context by comparing the records to

existing data from east Antarctic cores of EPICA Dome C (EDC), Vostok and central Dome Fuji. The data suggest that $nssCa^{2+}$ is contributing to the effective scavenging of $NO_3^-$ from the atmosphere through the formation of $Ca(NO_3)_2$. The geographic pattern implies that the process of $Ca(NO_3)_2$ formation occurs during the long-distance transport of mineral dust from the mid-latitude source regions by Southern Hemisphere Westerly Winds (SHWW) and most likely over the Southern Ocean. Since $NO_3^-$ is dust-bound and the level of dust mobilized through AIM events is mainly regulated by the latitudinal position of

SHWW, we suggest that $NO_3^-$ may also have the potential to provide insights into paleo-westerly wind pattern during the events.

## 1. Introduction

Nitrate ($NO_3^-$), the end product of oxidation of nitrogen oxides ($NO_x = NO+NO_2$) in the atmosphere, is one of the anions widely measured in Antarctic ice cores (Legrand and Mayewski, 1997; Wolff, 2013). A range of sources relevant to Antarctica has

been identified for the production and release of $NO_x$ species, such as a) oxidation of $N_2O$ and the photo-dissociation and



photoionization of $N_2$ in the stratosphere (Calisto et al., 2011; Savarino et al., 2007; Traversi et al., 2016)  b) thunderstorm lightning in the lower latitudes (Lee et al., 2014; Schumann and Huntriser, 2007), c) combustion of fossil fuels and biomass (Lee et al., 2014; Schumann and Huntriser, 2007), d) oceanic-sourced organic $NO_3^-$ (Beyersdorf et al., 2010; Wolff, 2013), and d) soil denitrification (Lee et al., 2014; Legrand et al., 1999; Schumann and Huntriser, 2007; Wolff, 2013). This variety

of sources makes the interpretation of $NO_3^-$ as an environmental proxy very interesting and highly complex. In addition, post-depositional losses through photolysis of $NO_3^-$, which involves its dissociation into NO, $NO_2$ and HCHO compounds (Erbland et al., 2013; Grannas et al., 2007; Shi et al., 2015) and volatilization of nitric acid ($HNO_3$) can also alter the original nitrate concentration preserved in the snow (Rothlisberger et al., 2002; Shi et al., 2019). The relative contribution of photolysis vs. volatilization also depends on the site characteristics, as the leakage through the photolytic process could be critical for the

high elevation inland sites where the snow accumulation rate is low (Erbland et al., 2013; Shi et al., 2015) and the light attenuation depth is deeper (Winton et al., 2020), whereas volatilization is found play a greater role in the warmer locations, with temperatures higher than ~ -24°C (Shi et al., 2019).

In general, the concentration of $NO_3^-$ is observed to be strongly influenced by temperature and accumulation rate, with $NO_3^-$ concentration found to be increasing with decreasing temperature (Rothlisberger et al., 2000; Rothlisberger et al.,

2002). This may be due to a higher uptake of $HNO_3$ into the snow with lower temperatures (Rothlisberger et al., 2000) or due to a higher stratospheric input associated with the sedimentation of polar stratospheric clouds (Grannas et al., 2007; Mayewski and Legrand, 1990). The role of accumulation rate in controlling the concentration of $NO_3^-$ can be difficult to decipher, as temperature and accumulation rate are interlinked (Rothlisberger et al., 2000). Nevertheless, an increase in the mean concentration of $NO_3^-$ is to be expected with higher accumulation rates because of reduced post-depositional leakage

(Rothlisberger et al., 2002; Zatko et al., 2016).

Interestingly, the concentration of $NO_3^-$ is instead observed to be higher during glacial times when the accumulation rate is low, a physical scenario where $NO_3^-$ loss is expected to aggravate due to enhanced post-depositional leakages (Rothlisberger et al., 2000; Wolff et al., 2010). Although the lower temperature can explain this relationship to some extent, it also necessitates additional mechanisms. This is because $NO_3^-$ shows significant millennial-scale variability within the glacial

period, as documented in the Vostok $NO_3^-$ record, when the temperature changes were minimal (~1 - 3°C) (Legrand et al., 1999; Rothlisberger et al., 2000). A recent ground-based study from Halley station, a site in coastal East Antarctica, shows only ~ 4 parts per trillion (ppt) variation in $HNO_3$ over a temperature change of ~10°C (Jones et al., 2014), suggesting that to have substantial adsorption or desorption of $HNO_3$ onto/out-of-the snow surface require a large temperature gradient, several times higher than what is observed during the glacial period.

Correlation of non-sea salt calcium ($nssCa^{2+}$), a proxy for mineral dust, and $NO_3^-$ over the glacial period in the cores EPICA Dome C (EDC), Vostok and Dome Fuji indicate that $nssCa^{2+}$ from dust particles react to form $Ca(NO_3)_2$ and stabilizes the $NO_3^-$ concentration (Legrand et al., 1999; Rothlisberger et al., 2000; Rothlisberger et al., 2002; Watanabe et al., 1999; Wolff et al., 2010). Whether this process primarily takes place in the atmosphere or the snow pack is still subject to much debate. If the reaction occurs in the snow, then mineral dust may be preventing the post-depositional loss of $NO_3^-$. If it takes



place in the atmosphere, then dust most likely enhances $NO_3^-$ scavenging (Rothlisberger et al., 2000). Rothlisberger et al. (2000) recommended that additional records for both mineral dust and $NO_3^-$ are required to resolve the role of dust in the accumulation of $NO_3^-$ during the glacial.

In this context, we present a new, well-dated and highly resolved glacial record of $NO_3^-$ and $Ca^{2+}$ from Roosevelt Island Climate Evolution (RICE) ice core, a record from Roosevelt Island, located in the eastern Ross Ice Shelf, West
Antarctica (Fig. 1). We focus on the temporal evolution of $NO_3^-$ and $nssCa^{2+}$ between 26 - 40 kilo years before present (ka BP), a time interval also characterized by millennial-scale climate oscillations in Antarctica, known as 'Antarctic Isotope Maxima' (AIM). We examine the response of the high-resolution record of $NO_3^-$ and $nssCa^{2+}$ in relation to AIM events, and on a range of time scales, and evaluate the implications of their association.

## 2. Materials and Methods

### 2.1 Site description

The RICE project is a 9-nation collaborative project that retrieved a deep ice core from Roosevelt Island, a local ice rise in Ross Ice shelf, West Antarctica (Bertler et al., 2018) (Fig. 1). The bedrock surface of the rise lies ~214 m below the sea level. The locally accumulated ice creates a dome of elevation of ~550 m above sea level and with a total thickness of about 764 m
at the crest. The drilling site (79°.364 S, 161°.706 W; annual mean temperature – 23.5°C) was chosen close to the dome summit. Annual average snow accumulation rates range between $22\pm4$ cm.w.e.a$^{-1}$ close to the drill site from 2010 - 2013 (Bertler et al., 2018; Winstrup et al., 2019).

### 2.2 RICE 17 Age Model

The RICE 17 chronology is a combination of annual layer counting for the top 343 m, covering the past 2700 years (Winstrup
et al., 2019) and gas synchronization for the remaining section of the core (Lee et al., 2020). For the time interval between 1971 CE to 30.6 ka BP, RICE $CH_4$ and $\delta^{18}O_{atm}$ profiles are matched to those of West Antarctic Ice Sheet (WAIS) Divide ice core (WDC) using an automated algorithm, and for the interval between 30.6 and 40 ka BP, a set of $CH_4$ and $\delta^{18}O_{atm}$ profiles in RICE were instead visually matched (Lee et al., 2020). A dynamic version of the Herron–Langway model has been used to simulate the Δ age and the RICE 17 ice-age scale is derived by adding the Δage to gas ages.

### 2.3 Sample Analyses

Major ions, such as $NO_3^-$, $Ca^{2+}$, $Na^+$, $Cl^-$, methane sulphonic acid (MSA), $SO_4^{2-}$, $K^+$ and $Mg^{2+}$ were measured using reagent-free Dionex ion chromatography (IC) system–5000, with a 2mm column, and a flow rate of 0.25 ml/min. Core processing and sample collection for IC and continuous flow analysis (CFA) of $Ca^{2+}$ is discussed in detail in Winstrup et al. (2019) and is based on a modified version of the Copenhagen CFA system (Bigler et al., 2011). For CFA calcium record, a three-point
calibration is used. The precision of the major ions analysed by IC is calculated based on the internal standards run regularly





between the samples on a ratio of 5:1 (sample: standard) The analytical precision is better than ~5 % for $Ca^{2+}$ and better than ~10% for $NO_3^-$ for the IC samples measured within the bracketing time period (26 - 40ka BP). A likely reason for the reduced precision observed for $NO_3^-$ could be due to the influence of extremely small contamination from drill fluid, a mixture of Estisol-240 and Coasol (Bertler et al., 2018), as the section of the core lies in the brittle ice zone (500 - 764m) that is susceptible
to internal fractures (Pyne et al., 2018).

### 2.4 Sample Analyses

Non sea salt fraction of calcium ($nssCa^{2+}$) is calculated using the linear equation (1)

$$nssCa^{2+} = R_c * ([Ca^{2+}] - R_m([Na^+]) * (R_c - R_m)^{-1}$$  (1)

where $R_c$ and $R_m$ are the crustal and marine ratio of $Ca^{2+}/Na^+$ (Lambert et al., 2012) and with $Ca^{2+}$ and $Na^+$ measured from IC.
We have used the traditional values of $R_c$ and $R_m$, which are 1.78 and 0.038, respectively (Bowen 1979), instead of 1.06 and 0.043 as used for $nssCa^{2+}$ calculation in EDC (Lambert et al., 2012). This is because sea salt aerosols deposited at RICE originate from open-ocean sources and are formed through mechanisms like bubble-bursting (Winstrup et al., 2019), as opposed to sites in central East Antarctica (e.g. EDC), where sea ice surfaces are identified as the major source of such aerosols (Wolff et al., 2006). Hence, traditional $R_m$ value shall be more appropriate. Also, the trajectory modelling for dust transport to
RICE shows a mixture of sources such as south of South America, Australia and New Zealand (Neff and Bertler, 2015), which suggests using a general value of $R_m$ may better represent the crustal ratio. However, it is also to be noted that changing $R_c$ and $R_m$ from traditional values has only a very minimal effect on $nssCa^{2+}$ calculation (mean difference of ~0.5ppb), as documented in numerous other studies (Bigler et al., 2006; Lambert et al., 2012).

### 2.5 Statistical Analyses

#### 2.5.1 Principal Component Analysis

Principal Component Analysis (PCA) is a multi-variate statistical technique used for data reduction and development of multi–parameter proxies of climate indicators (Buizert et al., 2018; Lambert et al., 2012). PCA analysis has been performed on the whole data set and is carried out using MATLAB algorithm, *pca*. Prior to the analysis, outliers are removed, data are averaged to 50-years to achieve equal sample spacing and are then detrended.

#### 2.5.2 Wavelet Coherence and Cross Spectrum Analyses

Wavelet coherence is a measure of correlation between two variables ($NO_3^-$ and $nssCa^{2+}$ here) in a time-frequency plane and is computed using analytic Morelet wavelet in MATLAB (Grinsted et al., 2004). The computed coherence is expressed as magnitude-squared coherence (msc). Wavelet cross spectrum (WCS) analysis is performed to identify the relative lead/lag between $NO_3^-$ and $nssCa^{2+}$ and the phase of WCS is provided for the values higher than 0.6 msc.






## 3. Results

### 3.1 Temporal Variability of nssCa$^{2+}$ and NO$_3^-$

We compare the changes in nssCa$^{2+}$ and NO$_3^-$ to RICE δD, a proxy for isotope temperature reconstructions (Dansgaard, 1964), between 26-40 ka BP (Fig. 2). Temperature records for the last glacial period, especially for marine isotope stage (MIS) 3 in

Antarctica are characterized by millennial-scale AIM events (Buizert et al., 2015; EPICA Community Members, 2006). RICE δD record also captures these oscillations and AIM events 3, 4, 5.1, 5.2, 6, 7, 8 are identified during this period (Fig. 2d). Among them, AIM 8 and AIM 4 are considered to be large events, in terms of both their duration (~ 2000 years) and degree of warming (~ 3°C) (EPICA Community Members, 2006).

RICE Ca$^{2+}$ has been measured independently using both CFA (Fig. 2a) and IC. The significant correlation between

both the Ca$^{2+}$ records (r=0.77, p<0.01; 50-yr averaged data), despite having been produced from two different analytical set up indicates the reproducibility of the data. Since the CFA system did not measure Na$^+$, we are not able to determine nssCa$^{2+}$ from CFA. Henceforth, here we use CFA Ca$^{2+}$ to complement the nssCa$^{2+}$ derived from IC, as Ca$^{2+}$ during this period is dominated by crustal inputs (average enrichment by a factor of ~2.7 from the marine ratio: Fig. S1).

The nssCa$^{2+}$ record shows a significant increasing trend towards MIS 2/last glacial maximum (LGM), with an increase

in mean concentration after 31.5 ka BP (~7.9 to 9.4 ppb) (grey line on Fig. 2b). During AIM events, nssCa$^{2+}$ signature broadly shows a characteristic response in the form of its fall in concentration (Fischer et al., 2007a; Lambert et al., 2012). However, during events AIM 6 and AIM 3, after the initial fall, the evolution involves a sharp rise in the concentration of approximately twice the magnitude of the fall, followed by a drop to the earlier level. This attributes a distinct signature to the dust signals of these events. In addition, the record also shows two large centennial-scale peaks around ~33.3 ka BP and 31.5 ka BP, which

coincide with the end of AIM events 6 and 5.2, respectively (black circles on Fig. 2b).

Similarly, NO$_3^-$ displays concentration decrease during AIM events (Fig. 2c). However, during AIM 6, we also observe a brief rise in the NO$_3^-$ concentration after the first drop, followed by its decline. Likewise in nssCa$^{2+}$, NO$_3^-$ record also includes two large peaks of centennial duration around 33.3 ka BP and 31.5 ka BP, respectively (black circles on Fig. 2c). The evolution of NO$_3^-$ over this period also showcase a shift to higher concentration after 31.5 ka BP (~ 22.5 to 24.4 ppb) (grey

line on Fig. 2c), which imparts a significant increasing trend to the record towards MIS2/LGM.

### 3.2 Wavelet coherence and cross spectral analysis of NO$_3^-$ and nssCa$^{2+}$

The statistically significant coupling between NO$_3^-$ and nssCa$^{2+}$ observed in their respective time series on millennial and centennial scales are also evident in the wavelet coherence and WCS analysis (Fig. 3a, b). A significant correlation is observed

in the frequency band of ~1500 - 3000 years between ~39 - 28 ka BP and within ~500 - 700 years between 30 - 35 ka BP. In the latter, centennial frequency band, WCS analysis also reveals an in-phase relationship (Fig. 3b).

### 3.3 Nature of association as revealed from Principal Component Analysis.





To understand the complex relationship between $NO_3^-$ and $nssCa^{2+}$, a principal component analysis is performed. The full data set consists of $Na^+$, $Cl^-$, $Mg^{2+}$, $nssCa^{2+}$, $K^+$, $SO_4^{2-}$, $NO_3^-$ and MSA concentrations. The first three principal components together explain ~95 % variance of the considered ions (Table 1). The first principal component (PC1) is dominated by the most common sea salt species, $Na^+$ and $Cl^-$, and accounts for 77.67 % variance of the data. The higher variability of sea salt aerosols in RICE is not a surprise as it is a low-elevation coastal site and, as such, will be very sensitive to any changes in the spatial configuration of the open ocean, especially Ross Sea (Bertler et al., 2018). The second principal component (PC2) explains 12.26 % variance of the data and is almost solely represented by $SO_4^{2-}$ (factor loadings=0.95). $SO_4^{2-}$ inputs to RICE can be a combination of biogenic contributions from Ross Sea (Winstrup et al., 2019) and volcanic emissions. Winstrup et al. (2019) have reported strong seasonality in $SO_4^{2-}$ signals in the top section of the RICE core (0-343m), with maximum concentration in summer when the phytoplankton activity increases, and minimum concentration in winter when the biological productivity in the Ross Sea is reduced. Volcanic $SO_4^{2-}$ has been widely used in Antarctic cores for developing common age scales (Parrenin et al., 2012; Severi et al., 2012) and reconstructing the past volcanic aerosol forcing (Sigl et al., 2014). The third principal component (PC3), which represents 4.94 % variance of the data, is characterized by the co-occurrence of $NO_3^-$ and $nssCa^{2+}$ with high factor loadings (0.84 and 0.45, respectively). Most of the variance of these two species are explained in this PC. As $NO_3^-$ and $nssCa^{2+}$ has completely different sources, this PC is most likely representing identical depositional mechanisms.

## 4. Discussion

### 4.1 Role of mineral dust in the $NO_3^-$ accumulation during the glacial period

Mineral dust, represented by the $nssCa^{2+}$, mostly originates from continents outside of Antarctica and is introduced into the Antarctic atmosphere by SHWW (Fischer et al., 2007b; Lambert et al., 2012; Röthlisberger et al., 2002). The strong association of $nssCa^{2+}$ and $NO_3^-$ observed in the time series, wavelet coherence and through their covariance in PC3, raises the question: Is $nssCa^{2+}$ an effective scavenging agent for the deposition of $NO_3^-$ or is the prevention of post-depositional leakage of $NO_3^-$ from the snowpack is the dominant influence? Rothlisberger et al. (2000) noted that if $NO_3^-$ formation takes place in the atmosphere, a strong correlation between $nssCa^{2+}$ and $NO_3^-$ is expected and is to be observed across the continent, as extra-continental dust is well-mixed in the Antarctic atmosphere by SHWW.

To test this hypothesis, we compare glacial $nssCa^{2+}$ and $NO_3^-$ records from other Antarctic locations (Fig. 1). Just as it is the case at our west Antarctic site RICE, EDC (Rothlisberger et al., 2000), Vostok (Legrand et al., 1999) and Dome Fuji (Watanabe et al., 1999), three east Antarctic sites, also document a similar close association between $NO_3^-$ and $nssCa^{2+}$ (Fig. 4). Glacial accumulation rate is relatively higher at RICE (~11$cmyr^{-1}$) (Lee et al., 2020), in comparison to the East Antarctic sites, where the rate of accumulation is understood to be very low during the glacial period (~1.4 $cmyr^{-1}$) (EPICA Community Members, 2006). Based on a global chemical transport modelling, Zatko et al. (2016) examined the impact of the snow accumulation rate on $NO_3^-$ loss, recycling and redistribution across Antarctica. Modelled $NO_3^-$ loss and recycling is very significant (up to ~ 95% and factor of 8, respectively) in the low accumulation polar plateau due to photolysis. This suggests that at Antarctic locations such as EDC, Vostok and Dome F, $HNO_3$ deposited on the snow surface are most likely to be lost





witout diffusing into the snow and reacting with $nssCa^{2+}$ present inside. Whereas, the photolytic $NO_3^-$ loss is minimal at the coastal Antarctic locations like RICE (5 - 55 %). Thus, as the $NO_3^-$ and $nssCa^{2+}$ association is found active at sites both in East

and West Antarctica that are spatially well-dispersed, with different rates of accumulation and hence preservation potentials, we propose that $nssCa^{2+}$ is scavenging $NO_3^-$ from the atmosphere and depositing it as $Ca(NO_3)_2$ in the snowpack.

We also note that overall during the glacial period, the hydrological cycle was weaker, continental shelves were more exposed and glacier discharges into outwash plains were higher, providing conditions conducive to increased dispersion of dust from the source regions (Fischer et al., 2007a; Lambert et al., 2012). Such dustier conditions can lead to more efficient

scavenging of $NO_3^-$. Studies have also shown that the lifetime of $HNO_3$ in the atmosphere through the dust removal can be well limited to just two days during dustier periods like glacial times (Hanish and Crowley, 2001), re-affirming its effectiveness in the scavenging process.

### 4.2 Environmental implications of $nssCa^{2+}$ and $NO_3^-$ association for the glacial


### 4.2.1 Mineral dust as proxy for SHWW during AIM events

Mineral dust, measured in the form of $nssCa^{2+}$, has been commonly used as a proxy to infer the latitudinal displacement of SHWW during the last glacial period, and particularly during AIM events (Lambert et al., 2012), where each event was accompanied by a poleward shift in the SHWW (Buizert et al., 2018; Markle et al., 2017). Southern South America is often

identified as the most important source for continental dust to Antarctic ice cores, especially to East Antarctica during the glacial period (Delmonte et al., 2008; Fischer et al., 2007b). However, some studies also indicate a possible contribution from Australia (De Deckker et al., 2010). Air mass trajectory and provenance studies analysis suggests Australia and New Zealand dust sources might contribute significantly to west Antarctic sites like RICE along with South American dust (Neff and Bertler, 2015; Winton et al., 2016). Evidence of widespread formation of loess deposits in New Zealand from glacier erosion in the

Southern Alps during the last glacial (Eden, 2003) would also contribute to strengthening the importance of this source area.

As SHWW shift southward, away from these source regions in the mid-latitudes, AIM events are characterized by a reduced $nssCa^{2+}$ concentration in the Antarctic ice core records (Fischer et al., 2007a; Lambert et al., 2012). The southward displacement of SHWW may also change the precipitation pattern in the predominant source regions such as Patagonia by reducing its strength (Boex et al., 2013; Moreno et al., 2012). Furthermore, in response to the atmospheric reorganization,

oceanic fronts re-arrange and shift southwards during AIM events, bringing warmer subtropical waters to the mid-latitudes (Barker et al., 2009; De Deckker et al., 2012). This causes sea surface temperature in the mid-latitudes to rise (Barker et al., 2009; Lamy et al., 2007), which then could trigger regional warming (Boex et al., 2013). This warming when combined with reduced precipitation in prominent source areas like Patagonia, can possibly lead to glacier melting (Boex et al., 2013) and formation of pro-glacial lakes which can act as dust-traps in such areas (Sugden et al., 2009), and also facilitate the vegetation

growth on surfaces previously covered by snow, thereby stabilizing them. Hence, these feedback processes can further reduce the dust-mobilization during the events.



RICE also captures a decrease in nssCa$^{2+}$ level broadly during all AIM events. However, during AIM 6 and AIM 3, we also observe centennial-scale peaks within the drop. To put our data into a wider context, we compared our records to the available EDC and EPICA Dronning Maud Land (EDML) data (Fig. S2). And during AIM 3, we observe a similar pattern at both the sites. The increase in nssCa$^{2+}$ flux after the initial drop is observed to be approximately twice the magnitude of the fall at both the locations. Fogwill et al. (2015) suggest that the obliquity minimum around ~28.5 ka BP may have resulted in circum-Antarctic sea ice growth, influencing SHWW and forcing it to shift northward with higher speed. This might explain the centennial-scale jump in nssCa$^{2+}$ concentration after the early drop during AIM 3. SHWW is likely to re-adjust later as the stadial conditions continue to be prevalent in the North Atlantic (Buizert et al., 2018) forcing SHWW to shift poleward, which could describe the following fall back in the concentration. During AIM 6, the pattern of the rise in concentration after the initial reduction observed in RICE is not visible at EDC or EDML. This questions the continent-wide representation of this dust signal and perhaps even suggest that its appearance is limited only to the Ross Sea region/West Antarctica.

### 4.2.2 Potential of NO$_3^-$ as an additional paleo-westerly wind proxy?

Experimental studies show that amongst calcium-containing minerals, carbonates such as calcite (CaCO$_3$) and dolomite (CaMg(CO3)$_2$) are the most reactive in the presence of nitric acid (HNO$_3$) in the atmosphere (Gibson et al., 2006; Krueger et al., 2004; Vlasenko et al., 2006). The reaction can lead to the formation of calcium nitrate (Ca(NO$_3$)$_2$) following reactions (2) and (3) below (Gibson et al., 2006; Krueger et al., 2004). In addition to the reaction with HNO$_3$, CaCO$_3$ and CaMg(CO3)$_2$ can also react with nitrogen pentoxide (N$_2$O$_5$) to form Ca(NO$_3$)$_2$ given by reactions (4) and (5) (Gibson et al., 2006)

$$CaCO_3 + 2HNO_3 \rightarrow Ca(NO_3)_2 + CO_2 + H_2 \qquad (2)$$

$$CaMg(CO_3)_2 + 4HNO_3 \rightarrow Ca(NO_3)_2 + 2CO_2 + H_2O + Mg \qquad (3)$$

$$CaCO_3 + N_2O_5 \rightarrow Ca(NO_3)_2 + CO_2 \qquad (4)$$

$$CaMg(CO_3)_2 + 2N_2O_5 \rightarrow Ca(NO_3)_2 + 2CO_2 + Mg(NO_3)_2 \qquad (5)$$

The mineral composition of both southern South American and Australian dust aerosols show the presence of calcium-containing minerals such as calcite and Ca-rich plagioclase (Radhi et al., 2011; Zárate, 2003), and thus the reactions (2)-(5) could occur in the atmosphere. Furthermore, the reactions (2)-(5) accelerate when the relative humidity increases, encouraging preferentially Ca(NO$_3$)$_2$ formation because of the hygroscopic nature of Ca(NO$_3$)$_2$ (Krueger et al., 2004; Mahalinganathan and Thamban, 2016; Vlasenko et al., 2006). Due to Antarctic's characteristic low relative humidity and shallow atmospheric boundary layer, it is less likely for neutralization reactions to occur locally (Mahalinganathan and Thamban, 2016). Investigation of nssCa$^{2+}$/ NO$_3^-$ association in the snow pit samples of coastal and inland East Antarctica suggests that binding





reactions most likely occur over the Southern Ocean during the long-range transport of $nssCa^{2+}$ due to a higher relative humidity over the region (Mahalinganathan and Thamban, 2016). During the glacial period, a time when relative humidity in the Antarctic atmosphere is significantly decreased compared to present day, we propose that neutralization reactions between mineral dust and $NO_3^-$ most likely occurred over the Southern Ocean. This suggests a possible link to long-range transport of

$nssCa^{2+}$ and $NO_3^-$ by SHWW and also implies that the $NO_3^-$ signal preserved through the glacial period is the dust-bound $NO_3^-$ dispersed by the westerly winds to the Antarctic atmosphere.

The amount of extra-Antarctic dust reaching Antarctica from the mid-latitude source regions during AIM events is primarily controlled by the zonal position of SHWW (Lambert et al., 2012). As $NO_3^-$ is dust bound and the variability of the dust concentrations across AIM events depends on the latitudinal movement of SHWW, it implies that $NO_3^-$ concentration is

indirectly regulated by SHWW. This opens-up the possibility of utilizing $NO_3^-$ as a westerly wind proxy for the AIM events. During AIM events, $NO_3^-$ exhibits a reduction in concentration similar to that of $nssCa^{2+}$ (Fig. 2b, c). In addition to the congruent response we observe in the time series, the wavelet analysis also reveals statistically significant coherence among these species in the periodicity of AIM events (~2000 years) (Fig. 3a, b). To our observation into wider context, we assessed $NO_3^-$ and $nssCa^{2+}$ records from EDC and Dome F across AIM events (numbered in Fig. 4). The identical response of both

species is evident in both the cores during large events (AIM 4 and AIM 8) (Fig. 4c-f). The relationship can also be identified for smaller events (AIM 3, 5.1, 5.2, 6,7) in EDC (Fig. 4c, d). However, at Dome F, lower temporal resolution (~200 years) act as a hindrance to identify the smaller events (duration ~1000 years). Moreover, the wavelet analysis also shows significant coherence between $NO_3^-$ and $nssCa^{2+}$ in the frequency domain of AIM events at both the locations (Fig. 5a, b). This shows that dust- $NO_3^-$ coupling identified during AIM events can also equally be observed in other regions of Antarctica which are located

distantly from RICE (e.g. EDC, Dome F). This indicates a continent-wide validity for this relationship, which also strengthens our argument of using $NO_3^-$ as a westerly wind proxy during AIM events.

To investigate the phasing between $nssCa^{2+}$ and $NO_3^-$ records at different sites, we performed WCS analysis. In RICE, the $nssCa^{2+}$ shows a lead of ~100 - 200 years in the frequency window of AIM events (~2000 years) over the whole period (~28 - 39 ka BP) (Fig. 3b). Whereas at Dome F, the records are in-phase from ~39 - 34 ka BP (Fig. 5b). While, starting from

~34 ka BP, $NO_3^-$ shows a lead of ~100 - 250 years. On the other hand, at EDC, both the records show an in-phase relationship over the whole interval (~26 - 39 ka BP) (Fig. 5a). This observation of spatially-varying phase relationship is intriguing. However, resolving it would require more detailed investigation perhaps on the of site characteristics, such as distance between the site and dust source and therefore the transport time for the aerosols or difference in the residence time along the transport route.


## 5. Conclusion

The correlation between $NO_3^-$ and $nssCa^{2+}$ in the RICE record provides additional evidence of the role of calcium-containing mineral dust on the scavenging of $NO_3^-$ through the formation of $Ca(NO_3)_2$. This process is particularly important during the glacial period when the dust load is higher by a factor of up to ~25 than today. This association has been observed previously





in the east Antarctic cores of EDC (Rothlisberger et al., 2000), Vostok (Legrand et al., 1999) and central Dome Fuji (Watanabe et al., 1999). Here, we show the same relationship for the first time for a coastal site in West Antarctica, thereby showing it is an Antarctic-wide phenomenon during the glacial. We conclude that $NO_3^-$ scavenging by mineral dust in the atmosphere is the dominant mechanism for the $NO_3^-/nssCa^{2+}$ correlation observed in ice cores and binding reactions most likely occur during the long-range transport from mid-latitudes to Antarctica by SHWW. During AIM events, the $NO_3^-$ variability depends on the

level of mineral dust mobilized from mid-latitude source regions, which inturn influenced by the latitudinal position of SHWW. For this reason, we also suggest that $NO_3^-$ may be a useful paleo-westerly proxy for the AIM events of the glacial period.

**Data Availability**

The following data are available in the supplementary material of the manuscript. RICE $Ca^{2+}$ (CFA), $NO_3^-$ (IC), $nssCa^{2+}$ (IC)

and the other Antarctic ice core data sets used. The RICE data will also be made available at PANGEA Data Publisher (ww.pangea.de) upon the acceptance for publication.

**Author contributions**

A.U.V and N.A.N.B. designed the project. R.L.P helped with the major ion measurements. H.A.K produced the CFA data.

A.U.V led the manuscript preparation and all the authors contributed to the conceptual discussions and the interpretations of the data.

**Competing interests**

The authors declare that they have no conflict of interest.


**Acknowledgments**

This work is a contribution towards the Roosevelt Island Climate Evolution (RICE) program, funded by national contributions from New Zealand, Australia, Denmark, Germany, Italy, the People's Republic of China, Sweden and the United States of America. The main logistic support was provided by Antarctica New Zealand (K049) and the U.S. Antarctic program. This

work has been funded by Ministry of Business, Innovation and Employment Grants through Victoria University of Wellington (RDF-VUW-1103, 15-VUW-131) and GNS Science (540GCT32, 540GCT12). A.U.V would like to thank the RICE drilling team of 2011 - 2013 for making the samples available for the study and GNS RICE PhD scholarship (2016 – 2019) for supporting the work. V.H.L.W was supported by a Rutherford Foundation Post Doctoral Fellowship administered by the Royal Society Te Apārangi.The Danish contribution to RICE was funded by the Carlsberg Foundation's North–South Climate

Connections project grant. The research also received funding from the European Research Council under the European Community's Seventh Framework Programme (FP7/2007-2013) ERC grant agreement 610055 as part of the Ice2Ice project and the European Union's Horizon 2020 research and innovation programme under grant agreement No 820970 and is TiPES contribution #54.



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



**Figures**

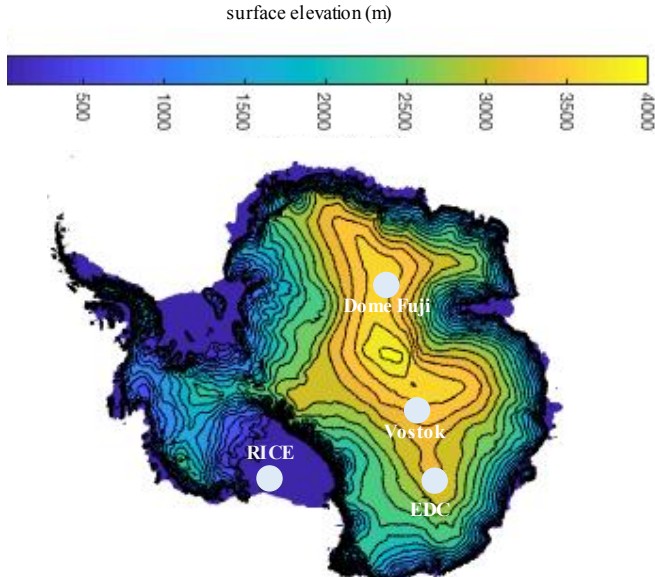

**Figure 1. Location Map of Roosevelt Island (RICE) on a Surface Elevation Map of Antarctica.** Deep cores where mineral dust and NO$_3^-$ correlation has been already documented are also shown - EPICA Dome C (EDC) (Rothlisberger et al., 2000; Wolff et al., 2010), Vostok (Legrand et al., 1999) and Dome Fuji (Watanabe et al., 1999). Map produced using Antarctic mapping tool box (Greene et al., 2017).

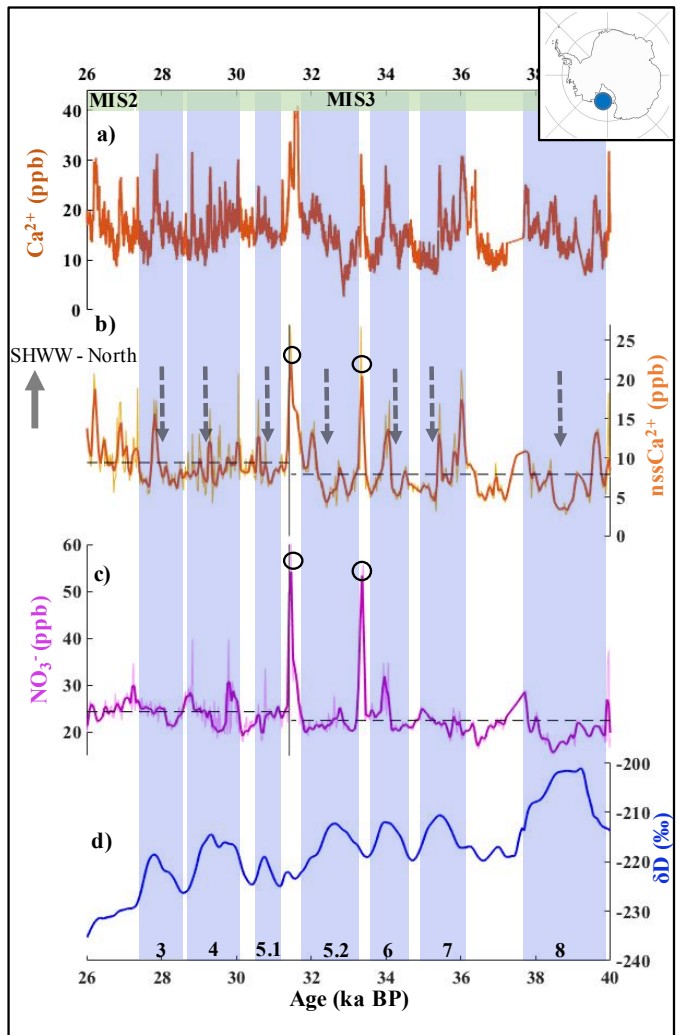

**Figure 2. Temporal Changes in RICE a) Continuous Flow Analysis (CFA) Ca$^{2+}$ b) nssCa$^{2+}$ c) NO$_3^-$ d) δD, between 26 - 40 ka BP.**
AIM events identified in RICE δD are shown in blue bars and are labelled at the bottom. Arrows pointing downwards indicate the lowering of the concentration in nssCa$^{2+}$ and NO$_3^-$ during AIM events. Black dashed lines in nssCa$^{2+}$ and NO$_3^-$ indicate the mean before and after the shift, demarcated by a grey line. Two large peaks in both the records are highlighted using black circles. Location of Roosevelt Island is shown in the inserted map, top right. Solid brown and pink lines indicates 50-yr average of the original nssCa$^{2+}$ and NO$_3^-$ data. δD record shown here is smoothed with a 500-yr box average (Lee et al., 2020). The latitudinal position of Southern Hemisphere Westerly Winds



(SHWW) based on nssCa$^{2+}$ concentration is indicated using an arrow on the left side, adjacent to nssCa$^{2+}$ plot. Marine isotope stages (MIS) covered by this time period are shown in green bars at the top.





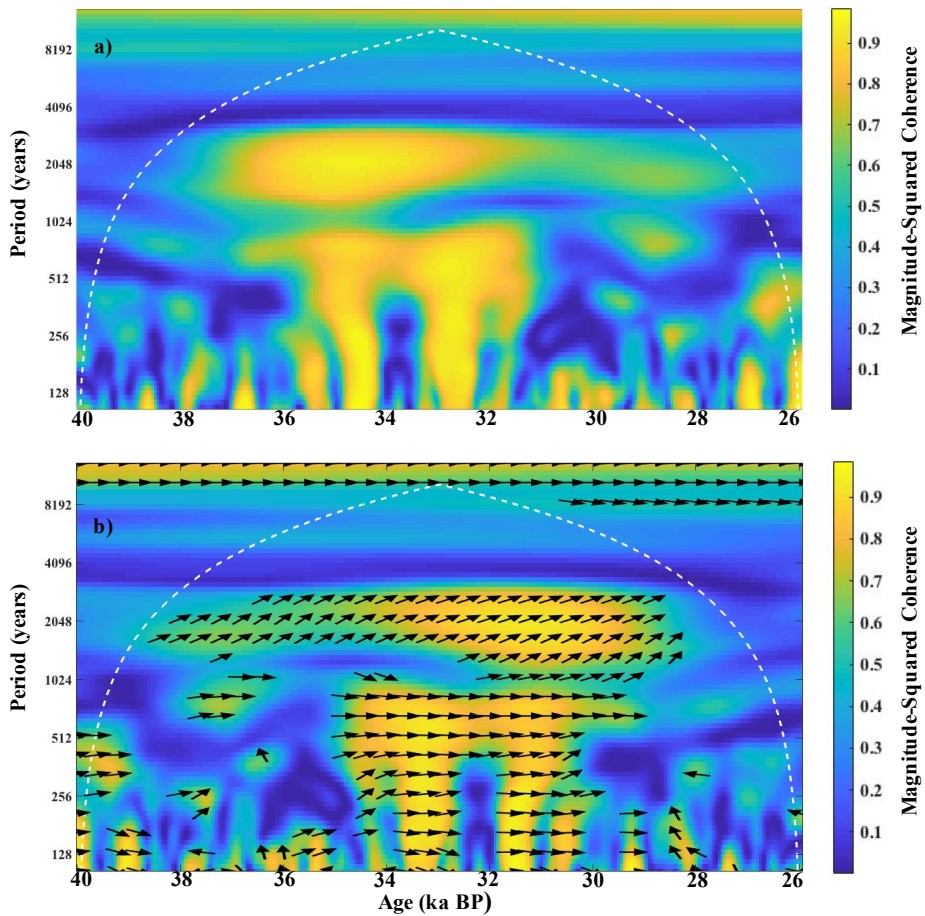

**Figure 3. Wavelet Coherence (a) and Cross Spectrum Analysis (b) of NO$_3^-$ and nssCa$^{2+}$.** The colours indicate the coherence values. The
cone of influence, where edge effect occurs, is shown in white dashed lines. Arrows in the cross-spectrum indicate the relative phasing
between nssCa$^{2+}$ and NO$^{3-}$ time series. A right pointing arrow corresponds to zero degree, i.e. both the series are in phase. An upward arrow
indicates a quarter cycle of lead of nssCa$^{2+}$ and a downward pointing arrow corresponds to a quarter cycle of lead of NO$_3^-$.


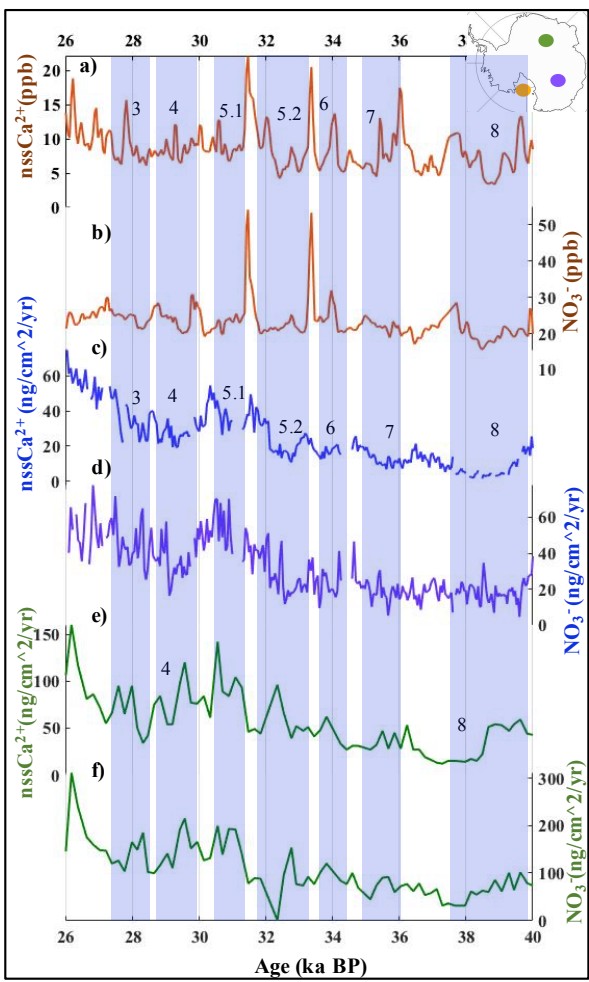

**Figure 4. Mid MIS 3 to MIS 2 Evolution of a) RICE nssCa²⁺(this study) b) RICE NO₃⁻(this study) c) EPICA Dome C (EDC) nssCa²⁺ (Fischer et al., 2007a; Rothlisberger et al., 2000b; Röthlisberger et al., 2002; Wolff et al., 2010) d) EDC NO₃⁻ (Rothlisberger et al., 2000b; Wolff et al., 2010) e) Dome Fuji (Dome F) nssCa²⁺ (Goto-Azuma et al., 2019; Watanabe et al., 1999) and f) Dome F NO₃⁻ (Goto-Azuma et al., 2019; Watanabe et al., 1999).** NO₃⁻ and nssCa²⁺ show close association at these locations. Blue bars indicate AIM events in RICE nssCa²⁺record. Events are identified in the nssCa²⁺records of all the three cores and are labelled. Due to a lower temporal resolution, only larger events are marked in Dome F. Geographic locations of the sites are shown in the inserted map. EDC data are plotted on Antarctic ice core chronology (AICC) 2012 age scale (Veres et al., 2013) and Dome F records are on DFO-2006 time scale (Kawamura





et al., 2007). Vostok is another core where $NO_3^-$-$nssCa^{2+}$ relationship has been previously documented, but not shown in the figure. Because of different age models, the timing of the events varies between the three sites.





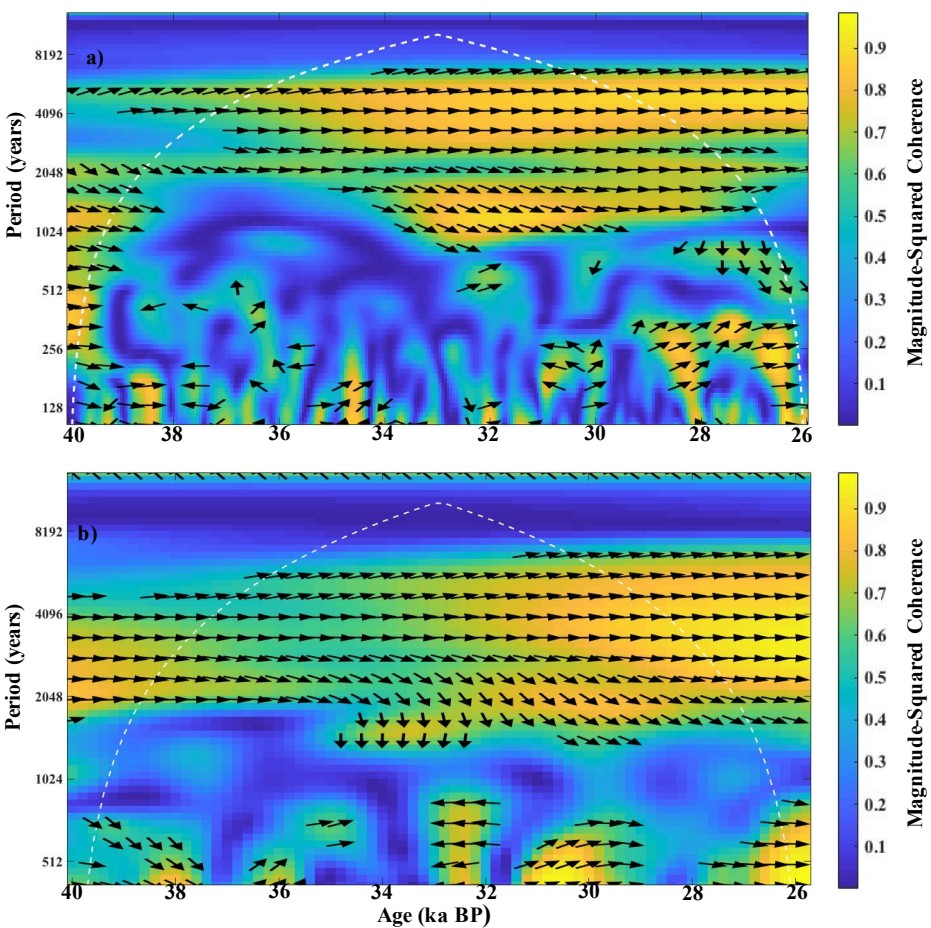

**Figure 5. Wavelet Cross Spectrum Analysis of a) EDC NO$_3^-$ and nssCa$^{2+}$ b) Dome F NO$_3^-$ and nssCa$^{2+}$.** Descriptions are same as in that of Fig. 3. Prior to the analysis, data sets are sampled to equal intervals (50 - yr in EDC and 200 - yr in Dome F).



**Tables**

| Ions | PC1 | PC2 | PC3 | PC4 | PC5 | PC6 | PC7 | PC8 |
|---|---|---|---|---|---|---|---|---|
| Cl⁻ | *0.84* | -0.06 | -0.13 | -0.49 | 0.02 | 0.04 | -0.08 | -0.01 |
| $NO_3^-$ | 0.03 | 0.06 | *0.84* | -0.14 | -0.36 | 0.33 | -0.02 | -0.10 |
| $SO_4^{2-}$ | 0.12 | *0.95* | -0.05 | 0.11 | -0.11 | -0.15 | -0.11 | 0.002 |
| MSA | 0.05 | 0.11 | -0.07 | 0.24 | 0.49 | 0.78 | -0.25 | 0.009 |
| Na+ | *0.50* | -0.17 | 0.16 | 0.80 | -0.07 | -0.15 | 0.12 | -0.009 |
| $K^+$ | 0.01 | 0.0004 | 0.15 | -0.02 | 0.04 | -0.03 | -0.05 | 0.98 |
| $Mg^{2+}$ | 0.04 | 0.17 | 0.07 | -0.09 | 0.29 | 0.12 | 0.92 | 0.02 |
| $nssCa^{2+}$ | 0.003 | 0.01 | *0.45* | -0.07 | 0.71 | -0.45 | -0.21 | -0.13 |
| **%Variance** | **77.67%** | **12.26%** | **4.94%** | 2.74% | 1.17% | 0.70% | 0.42% | 0.05% |

**Table 1. Results of the Principal Component Analysis**. The first three principal components (PC1, PC2 and PC3) are identified as significant and their variance is highlighted in bold. Factor loading of dominant species in PC1, PC2 and PC3 are highlighted in bold italic.
