# Peer review of "Mineral Dust Influence on the Glacial Nitrate Record from the RICE Ice Core, West Antarctica and Environmental Implications"

_Climate of the Past, 2020_

## Referee Comment (RC1) · Anonymous Referee #1 · 30 Dec 2020

This manuscript concerns the relation between the nssCa2+ and NO3- records obtained from the RICE ice core in coastal West Antarctica. The idea behind the manuscript is to shed light on the correlation between these two proxies and in particular on the stability effect that nssCa2+ seems to have on nitrate preservation in Antarctic ice.

The topic is of great interest for the ice core community and more in general for the paleoclimate one, since the interpretation of Antarctic NO3- records is not yet fully constrained and the significance of this proxy has to be improved. The authors show a good knowledge of literature and the reading of their work is good. But despite these

positive points I admit I have some major concerns for some of the points presented and discussed by the authors and for this reason I cannot support the publication of the manuscript in its current form. Major revision should be done in order to re-consider this work for publication.

My major concern is the accuracy of the presentation of results and some key aspects of the interpretation. For example, I find the descriptions of both nssCa2+ and NO3- records, not really accurate and constrained (see below my specific comments). Most of the discussion relies on the fact that for the authors the two proxies are well correlated in their records. To be honest I don't find a strong correlation, as the authors repeat several times along the manuscript. If I look at bare records (figure 2), I see that most of the oscillations of the isotopes, are not accompanied by similar (or even opposite) oscillations in the nssCa and nitrate records. There are two major events showing a peak in nssCa and NO3, which are well highlighted by wavelet analysis. But if I consider the entire record, I cannot appreciate such a strong correlation.

The authors should apply more robust methods to discuss about the correlation. At first, I would prepare a simple scatter plot between the two variables. This would give to the reader a general an immediate impression about the degree of correlation between nssCa and nitrate. After this I would pass to more sophisticated tools such as PCA and wavelet analysis.

Another important point discussed in the text is the increase of the mean of both nssCa and nitrate at about 31 ka BP. The authors don't provide any exhaustive explanation about the significance of such change and about the choice to place the change point at that age. Further details are found below.

I guess that an error is present in figure 3, where the authors present the results of wavelet analysis. It seems that the two figures are one the opposite of the other, as if one of them is presented backwards, I am not sure but this could have some consequences on the interpretation (see specific comments below).

About the definition of new proxies to investigate SHWW: looking at Figure 2 it is really hard to capture a correlation between nssCa and isotopes during AIMS. It is well known that dust concentration drops in Antarctic ice during cold stages but looking at Figure 2 I cannot appreciate this as it happens for example with the EPICA Dome C record. The difference is probably related to the influence of more local signals at Roosevelt Island compared to inner Antarctica, or to the role played by the Ocean. In the light of this, I would prefer if the authors would remove the part of the manuscript about the use of nssCa as a proxy for SHWW. Similarly, they should remove the part dedicated to the use of NO3- as an additional proxy for this. If they don't provide more stringent evidences about the significance of their findings, it will be better to focus the manuscript on the bare correlation between the two proxies and not also on their environmental significance.

My suggestion is to completely remove the part dedicated to the new proxies and focus all the attention on the investigation between nssCa, nitrate and isotopes. Giving more attention to the comparison with other sites. For example, they could prepare a new comparison where they compare a scatter plot (nitrate vs. nssCa) prepared for each considered ice core, providing the coefficients of determination or other tools to avaluate the degree of correlation. I guess that interesting facts could came out. . . Also an accurate change point analysis could bring interesting results.

About the choice of the time interval: you have decided to focus on MIS3 probably to give attention to AIMS. I do not completely agree with this decision. I would have extended the period so as to keep into account also the shift from MIS2 to Holocene, since also this major climatic shift have a strong impact on the ice concentration of both nssCa and nitrate. Extending the manuscript to keep into consideration this period would make it more attractive for the ice core community. Please consider the possibility to do this.

Another important thing: the authors always compare RICE to Dome C, Dome Fuji and EDML as if these sites have more or less the same characteristics. I guess that

the peculiar position of RICE should be more evidenced. What about adding another West Antarctic site to compare? For example WAIS? Since I see evident differences between RICE and the other East Antarctic sites, I am wondering if WAIS would be more similar to RICE, providing some additional clues. . .

Last but not least: did you consider the possibility that a fraction of the dust deposited at Roosevelt Island is local, mostly from the Dry Valleys? This would have some consequences on your interpretation (and would also explain why nssCa at RICE responds differently to climate changes if compared to inner antarctic sites).

More specific comments

Line 57: "shows only $\sim$ 4 parts per trillion (ppt) variation in HNO3 over a temperature change of $\sim$10°C" It is not clear if this variation refers to the atmosphere or to snow concentration

Line 69: not to repeat record two times in the same sentence, what about changing "a record from Roosevelt Island" with "a paleoclimatic archive from Roosevelt Island"?

Line 89: please add some information about the resolution of the core in the time interval considered in this study

Line 96-100: you say that precision of your measurements is about 5% for Ca and 10% for NO3-, you also justify this saying that probably a higher blank is present for nitrate because of the influence of the drilling fluid. I don't understand this passage from the analytical point of view. If you have a stronger background for NO3-, the detection limit should be affected at a first instance, not analytical precision. Maybe with NO3- you have on average a lower sample to noise ratio and for this reason your dispersion (linked to precision) is higher? Please better explain this. Since you talk about precision, what about accuracy? Do you have some information? Another thing that is not clear is the number of sampled you measured. Data presented in this study have been obtained through CFA or discrete sample analysis? In the second case,

what is the resolution of the measurements? You should clarify a little bit these points.

Line 118: what method did you apply to recognize and remove the outliers? Please give some details

Line 129: "especially for marine isotopic stage (MIS) 3,"

Line 131: replace "also" with "well"?

Line 134-135: what about presenting in a scatter plot this pair of variables: Ca2+ from CFA vs. Ca2+ from IC? This would be a nice supplementary figure graphically showing the correlation between the two independent measurements. Probably you should sample the data to allow for the comparison, but I guess it would be an interesting information to show.

Line 136-138: it is not well clear to me how you calculated nssCa from CFA using the data from IC. The way you coupled the data from the two techniques is not clear throughout the paper, this point should be improved. A suggestion: why don't move this section which is dedicated to the description of how you determined the various records (lines 134-139), in the Methods paragraph? It would make easier to understand how you prepared your records and keep the results distinct from the methodological aspects of the paper.

Line 139-140: I feel that defining this as a significant increase is not really supported by your data. A change of less than 2 ppb between values presenting a high variability is not so strong. To claim that this is a significant change the authors should present some more robust evidence based on statistical analysis, for example the application of a test to infer about the difference of mean values or an algorithm to detect change points in time series.

Lines 140-144: the anti-correlation between dust and isotopes is well known and discussed for inner Antarctic sites, both in relation to major climate changes and to millennial variability, as in the case of AIMs. This is correctly reported by the authors.

Looking at Figure 2 my impression is that at RICE things are different. In fact, I cannot see any significant decrease of nssCa associated with the occurrence of a AIM. On the contrary, in some cases there is a positive correlation, as highlighted by the authors: this is the case for AIM 3,6 and partially 7, where a peak of the isotopes corresponds with a peak of nssCa. The arrows highlighting the relative minima in the nssCa2+ records during AIMs (figure 2) are a little bit misleading. In most cases they highlight relative minima, but considering the same AIM event it is possible to find other relative maxima... Considering all these points, I must admit that it is difficult to follow this part of your manuscript because what observed at RICE doesn't look like to what observed in inner Antarctica. This is probably related to the peculiar geographic position of Roosevelt Island, where the atmospheric dust cycle is likely different from inner Antarctica. This part should be improved, data should be presented with more accuracy and be supported by statistical evidences, with a deeper comparison with inner Antarctica (and maybe other coastal sites?).

Line 149: also in this case I would appreciate if such a difference of the NO3 record would be investigated through a fully statistical approach.

Line 152-156: as anticipated, I have noticed that figure 3a and b are not exactly the same, one is backward of the other. This probably means that one of them is not shown properly, which one? Please fix this problem which I am afraid will affect the consequent discussion (in particular the direction of correlation). I believe that Figure 3a is the correct one, while 3b is presented backward. I have inferred this from the two evident correlation spikes at about 34.5 and 33 ka BP which corresponds to the two characteristic events highlighted with the black circles in figure 2. This should also be mentioned here.

Line 173: change to "is most likely related to a common depositional mechanism"

Line 178: looking at wavelet results I do find some correlation between the two variables, not enough to define it as "strong". The main reason is that the correlation is

high in particular between 36 and 32 ka BP, not along the entire time interval considered here. I would remove the term "strong".

Line 181: change "is the dominant influence" with "the dominant process"

Line 191: 85% and 8, it is not clear what these numbers refer to

Line 194: 5-55 %, also the meaning of these values is obscure

Line 215: here you should cite the recent paper by Koffman and colleagues (New Zealand as a source of mineral dust to the atmosphere and ocean)

Line 216-226: this sounds like an introduction, not as a discussion, if the main point of this part is that during AIM dust emission is suppressed, just say this with the appropriate reference.

Line 229-230: as already pointed out in my general comments, I guess that the use of nssCa at RICE as a proxy for SHWW is not well constrained by your results and interpretation. Also the identification of peaks similar to the ones found in RICE during AIM 3 and 6 at Dome C and Dronning Maud Land should be removed: the peaks at these two sites are not statistically significant and their anomaly perfectly falls within the spreading of data.

Line 240-290: since I don't believe that the evidences presented in this manuscript supports the possibility to use nssCa as a proxy for SHWW, I also believe that the same is not possible for Nitrate. For this reason I have not reviewed this part which, in my opinion, should be completely changed.

Best regards
* * *

---

## Referee Comment (RC2) · Anonymous Referee #2 · 18 Feb 2021

This manuscript presents new ionic chemistry data from the RICE ice core in the Ross Sea region of Antarctica and focuses on the relationship between nitrate and non-sea-salt calcium (nssCa). Data from MIS 3 are presented, encompassing AIM events 3-8. The authors claim that nitrate and nssCa co-vary across AIM events, as has been observed previously in 3 East Antarctic cores. This apparent similarity between RICE and the 3 East Antarctic cores is then assumed to be a continent-wide signal and stated to demonstrate that NO3 reacts with nssCa in the atmosphere prior to deposition (rather than in the snowpack). Finally, this potential interaction of nitrate and nssCa in the atmosphere is used as a basis to claim that nitrate could be used as a proxy for Southern Hemisphere Westerly Winds (SHWW) strength/position during the Last

Glacial.

I find the conclusions of this study uncompelling. They are barely supported by the data analysis presented. The statistical analysis itself is poorly explained and error-prone, meaning the critical observation of co-variation between nitrate and nssCa is placed in serious doubt. Going further, to extrapolate one coastal site to the entire Antarctic continent is not well-justified so we don't really learn anything about the controls on ice core nitrate. The claim of a new SHWW proxy should be removed entirely. In short, I cannot recommend this work for publication.

I provide specific comments below but I stopped at section 4.2.2 when it became clear the discussion had moved beyond what is justified by the data and analysis presented.

Major comments Time interval chosen: The authors don't state why this time interval was chosen for analysis. Why is the deglaciation not included? It would be interesting to see how the proposed nitrate-nssCa relationship responds under different accumulation rates. L197-202 hint at this but no Holocene data is presented so this text is redundant.

Data analysis: The primary observation of the paper is that nitrate and nssCa co-vary across MIS3. The first problem is that this is extremely difficult to see on Figure 2. Yes, there are two peaks (Circled on figure) that are coincident in the two records but that is about it. Maybe plotting the records on a log scale and closer together would help, but I am doubtful. There is no correlation reported between the datasets, despite the claim of "statistically significant coupling" made at L153. The second problem here is that variability in nssCa or nitrate linked to AIM events (as claimed L140) is near-impossible to see – the variability in both is much higher frequency.

Next is the observation that there is a systematic shift in nitrate and nssCa records at ∼31 ka. How was this change-point in the records identified? What is the significance of this observation – maybe I missed it by not reading through to the end.

It seems that the authors try make up for the lack of [reported] correlation between nssCa and nitrate by using more sophisticated statistical methods: PCA and wavelet analysis. I'm afraid I have concerns about the validity of both. PCA: More information about the pre-treatment of data is required. How are outliers defined? Are all the data mean-centred and normalised? PCA is not usually suitable for chemistry data that has a skewed distribution – it is often appropriate the log-transform data first to ensure you start with something like a normal distribution. The risk of not following these steps is that outliers heavily influence the result. Without seeing the datasets used or the resulting EOFs, it is difficult to know, but my concern is that PC1-3 are dominated by 'extreme' variability, e.g., the two coincident spikes in nitrate and nssCa in the case of PC3.

Wavelet analysis: I am not an expert on this technique but know a little. The text to describe both the method (L121-124) and the results (L153-156) is imprecise. The figures need a significance level adding, otherwise the results are meaningless. On Figure 3, the result is again dominated by the two high peaks in nitrate and nssCa. This adds little to the study.

Extrapolation of RICE result: The comparison of RICE to the 3 East Antarctic sites is almost redundant. We already know that nitrate and nssCa are related at these sites. But the RICE results alone aren't enough the infer something about the controls on nitrate (even if the analysis is robust). WAIS Divide nitrate and calcium data are available in high resolution for this time interval. I'm not aware that anyone has looked into the nitrate – your study would be strengthened if supporting evidence could be gleaned from another warm, West Antarctic (though not so coastal) core. https://www.usap-dc.org/view/dataset/601008

Minor comments - Did you consider comparison with acidity or DEP data, following Röthlisberger et al., JGR 2000. - There is no accumulation rate information included. Could it explain some of the variability in nssCa and/or nitrate? - L190-196: Doesn't N and O isotopic data from nitrate suggest a lot of coastal nitrate is actually recycled

from the interior? Sorry I can't remember the reference but it would be from Becky Alexander's group.

Figures: Fig. 1: Just a suggestion but could you label the elevation contours and include accumulation rate field in colored shading (even if it is present-day rather than glacial)? Accumulation rate is important for the nitrate preservation. Fig. 2: Inset map not needed if Fig.1 included. Fig. 3: As other reviewer commented, one plot appears to be flipped. Check colorbar labels are accurate.